# Alternate Mapping Correlated k-Distribution Method for Infrared Radiative Transfer Forward Simulation

**Feng Zhang** [1,2,*] , **Mingwei Zhu** [1] , **Jiangnan Li** [3] , **Wenwen Li** [1] , **Di Di** [4] , **Yi-Ning Shi** [1]
**and Kun Wu** [1]

[1] Key Laboratory of Meteorological Disaster, Ministry of Education (KLME)/Collaborative Innovation Center on Forecast and Evaluation of Meteorological Disaster (CIC-FEMD), Nanjing University of Information Science and Technology, Nanjing 210044, China; zmw_1994@nuist.edu.cn (M.Z.); liwenwen0213@nuist.edu.cn (W.L.); 578460165syn@sina.com (Y.-N.S.); wukun0820@sina.cn (K.W.)
[2] State Key Laboratory of Severe Weather, Chinese Academy of Meteorological Sciences, Beijing 100081, China
[3] Canadian Centre For Climate Modelling and Analysis, Science and Technology Branch, Environment Canada, Victoria, BC V8W 3P2, Canada; Jiangnan.Li@canada.ca
[4] Chinese Academy of Meteorological Sciences, China Meteorological Administration, Beijing 100081, China; didi13@mails.ucas.ac.cn
[*] Correspondence: fengzhang@nuist.edu.cn

**Abstract:** The alternate mapping correlated k-distribution (AMCKD) method is studied and applied to satellite simulations. To evaluate the accuracy of AMCKD, the simulated brightness temperatures at the top of the atmosphere are compared with line-by-line radiative transfer model (LBLRTM) or the observed data which are from Advanced Himawari Imager (AHI) on board the Himawari-8, as well as Medium Resolution Spectral Imager (MERSI) on board the Fengyun-3D. The result of AMCKD is also compared with the algorithm of Radiative Transfer for the Television Observation Satellite Operational Vertical Sounder (RTTOV). Under the standard atmospheric profiles, the absolute errors of AMCKD in all longwave channels of AHI and MERSI are bounded by 0.44K compared to the benchmark results of LBLRTM, which are more accurate than those of RTTOV. In the most cases, the error of AMCKD is smaller than the NEDT at ST, while the error of RTTOV is larger than the instrument noise equivalent temperature (NEDT) at scene temperature (ST). Under real atmospheric profile conditions, the errors of AMCKD increase, because the input data from ERA-Interim reanalysis dataause bias in the satellite remote sensing results. In the most considered cases, the accuracy of AMCKD is higher than RTTOV, while the efficiency of AMCKD is slightly slower than RTTOV.

**Keywords:** AMCKD; line-by-line; remote sensing; brightness temperature

## 1. Introduction

The computational speed of radiative transfer for satellite and ground based remote sensing measurementsis always a big issue [1–4]. The most precise way to simulate the atmospheric radiative transfer process is the line-by-line (LBLRTM) radiative transfer method, but its time-consuming nature prevents it from being applied to practical problems. For the new generation of satellite instruments providing high spatial and spectral resolution spectra, the correlated k-distribution (CKD) could be an appropriate method. In CKD, the gaseous absorption within a spectral interval is entirely unrelated to the variation of the absorption coefficient for wavenumbe and is contingent only on the distribution of k within the interval [5–8]. Gaseous transmission in the CKD model is integrated over a smooth and monotonically increasing absorption coefficient space instead of over a tortuously variable frequency space. Therefore, the CKD method requires many fewer points to calculate the spectral transmissivity than the LBLRTM, with an increase of computational efficiency by two to

three orders of magnitude. The traditional CKD method maps single gas within the whole band spectral region and then the same mapping is applied to the other gases, leading to a highly accurate result in the mapped gaseous absorption but relatively low accuracy to the other non-mapped gases. Alternate mapping correlated k-distribution (AMCKD) proposed in [9] is an efficient and precise CKD method, which takes the importance of all gases within the entire band into consideration. According to AMCKD, the cumulative probability space is divided into several intervals and then one primary gas is selected to sort in each interval, while the same sorting rule is applied to the other gases in this interval. AMCKD has been successfully applied in climate models and achieved very accurate results in climate simulation [10,11] .

For remote sensing, the forward radiative transfer model is used to simulate the reflectance/brightness at the top of atmosphere (TOA) for different satellite viewing zenith angles, which requires more overall accuracy and computational efficiency. In the last two decades, several radiative transfer models (RTM) have been developed for satellite simulation (e.g., Interpolation and Profile Correction (IPC) [12], k Compressed Atmospheric Radiative Transfer Algorithm (kCARTA) [13,14], σ-Infrared Atmospheric Sounding Interferometer (σ-IASI) [15], Radiative Transfer for the Television Observation Satellite Operational Vertical Sounder (RTTOV) [16]). However, AMCKD has seldom been applied to these models. The critical point in handling gaseous transmission in satellite simulation is to deal with the overlap of two or more gases, which is the same as the radiative transfer process in climate modes. Since AMCKD was designed to efficiently deal with the multiple gas transmission, it is natural to consider the application of AMCKD to satellite simulation and find out the consequences in accuracy and efficiency by comparing with other algorithms. The purpose of this study is to investigate this problem.

In Section 2, the theory of AMCKD is introduced, which is based on [9]. Furthermore, the algorithm is modified to fit the applications in satellite simulation. Section 3 introduces the simulation results of AMCKD, which are compared with those based on the popular radiative transfer algorithm RTTOV. In Section 4, we draw some concluding remarks.

## 2. Methods

In this section, we introduce how to apply AMCKD to satellite simulation. To include the effect of the spectral response function (SRF, $\phi(v)$) of the measurement instrument in the spectral transmissivity, gaseous transmittance of a single gas in a homogeneous layer of the atmosphere over the spectral interval is defined as [17–20]

$$Tr_\phi(u) = \frac{1}{F_{]\phi}} \int_{\Delta v} \phi(v) e^{-uk(v)} dv, \tag{1}$$

where $Tr_\phi$ is the gaseous transmittance, $u$ is the gas amount, and $k(v)$ is the gaseous absorption coefficient at wave number $v$. The average transmission can also be written in $k$ space [17] as

$$Tr_\phi(u) = \int_0^\infty e^{-uk} f_\phi(k) dk, \tag{2}$$

where $f_\phi(k)$ is the probability distribution function weighed by spectral response function for the gaseous absorption coefficient. Therefore, $f_\phi(k)$ is defined as

$$f_\phi(k) = \frac{1}{F_\phi} \int_{\Delta v} \phi(v) \delta[k - k(v)] dv. \tag{3}$$

$f_\phi(k)dk$ is the fraction $dv/\Delta v$ of the integration range in which the absorption coefficient weighed by spectral response function lied between $k$ and $k + dk$ Please note that $k$ space and $v$ wavenumber space could transform each other as

$$
\begin{aligned}
Tr_\phi(u) &= \int_0^\infty e^{-uk} f_\phi(k)dk \\
&= \int_0^\infty e^{-uk} \frac{1}{F_\phi} \int_{\Delta v} \phi(v)\delta[k - k(v)]dvdk \\
&= \frac{1}{F_\phi} \int_{\Delta v} e^{-uk(v)} \phi(v)dv
\end{aligned}
\tag{4}
$$

Consider the cumulative probability function defined as

$$
g(k) = \int_0^k f_\phi(k')dk',
\tag{5}
$$

Based on (2), the transmittance integral can also be expressed as

$$
Tr_\phi(u) = \int_0^1 e^{-uk(g)}dg,
\tag{6}
$$

As such, $g$ forms a cumulative probability space (CPS), and $k(g)$ is the absorption coefficient in CPS. Scaling CPS into $N_g$ points $G_i$ ($i = 1, 2, 3, \ldots, n$), with $G_0 = 0$, $G_{N_g} = 1$, and letting $g_i = G_i - G_{i-1}$, with $\sum_{i=1}^{N_g} g_i = 1$, the integral in (5) becomes

$$
Tr(u) = \sum_{i=1}^{N_g} \int_{G_{i-1}}^{G_i} e^{-uk(g)}dg,
\tag{7}
$$

The transmission function in AMCKD model in an approximation format can be calculated by

$$
Tr(u) = \sum_{i=1}^{N_g} e^{-u\langle k(g_i)\rangle} g_i,
\tag{8}
$$

where $\langle k(g_i)\rangle$ is the averaged absorption coefficient in $i$-th subinterval. We can obtain the averaged absorption coefficient $\langle k(g_i)\rangle$ from the line-by-line results in the same domain for a suitable range of $u$. As the inequality relation is given by

$$
\int_{G_{i-1}}^{G_i} e^{-u \cdot k(g)}dg \geq e^{-u \int_{G_{i-1}}^{G_i} k(g)dg},
\tag{9}
$$

which indicates that we need the adjusted mean absorption coefficient $\langle k(g_i)\rangle = \frac{\alpha_i}{g_i} \int_{G_{i-1}}^{G_i} k(g)dg$ to satisfy with

$$
\int_{G_{i-1}}^{G_i} e^{-u \cdot k(g)}dg = e^{-u\langle k(g_i)\rangle},
\tag{10}
$$

where the adjusted factor $\alpha_i \leq 1$. In climate models, $\alpha_i$ is pre-calculated through the comparison of transmittance with line-by-line results [9].

In the realistic inhomogeneous atmosphere, $k(g_i)$ becomes $k(g_i, P, T)$. AMCKD method assumes that the ordering of the strengths of absorption lines is the same as those of different temperature $T$

and pressure $P$ levels. $k(g_i, P, T)$ can be parameterized as a function of temperature polynomial for multiple reference-pressures

$$k(g_i, P_r, T) = \sum_{l=0}^{4} c_l(g_i, P_r)(T - 250)^l, \tag{11}$$

where $P_r$ is reference-pressure, and $c_l$ is the fitting coefficient. A total of twenty-one pressure levels are distributed between 1000 and 0.1 mb. Equation (12) is valid for temperatures between 160 K and 340 K, which includes most atmospheric conditions [9]. Then $k(g_i, P, T)$ at any arbitrary pressure $P$ can be obtained by the linear interpolation between two neighboring $P_r$ and $P_{r+1}$ ($P_{r+1} \leq P \leq P_r$) through Equation (11).

$$\begin{aligned} k(g_i, P, T) \quad &= \frac{P_r - P}{P_r - P_{r+1}} \cdot [k(g_i, P_{r+1}, T) - k(g_i, P_r, T)] \\ &+ k(g_i, P_r, T) \end{aligned} \tag{12}$$

The gaseous overlap is the key issue for any CKD algorithm [8]. AMCKD is based on the rearrangement of the gaseous absorption coefficient in each portion of CPS to capture the main characteristics of each gas. The CPS space is split into several portions; in each potion, a primary gas is chosen and sorted, and the same sorting rules are applied to other gases. An example is given in Figure 1, which shows the treatment of gaseous overlap based on AMCKD for a 9.6347 μm channel of Advanced Himawari Imager (AHI) centered at the wavelength of 9.6437 μm. The half-width of the 9.6347 μm channel of AHI is 40.73 cm$^{-1}$ (Figure 1a). The spectral absorption coefficients of $H_2O$, $CO_2$, and $O_3$ are shown in Figure 1b. The absorption coefficient of $O_3$ is stronger than that of the two other gases, but with less fluctuation. Figure 1c illustrates how to use AMCKD to deal with the multi-gas overlap. In the 9.6347 μm channel, the cumulative probability space is split into six intervals, with [0, 0.48] sorted by $H_2O$, [0.48, 0.77] sorted by $O_3$, [0.77, 0.875] sorted by $CO_2$, [0.875, 0.94] sorted by $H_2O$, [0.94, 0.973] sorted by $CO_2$, and [0.937, 1] sorted by $O_3$, respectively. The sorted absorption line becomes smooth and monotonous, while the absorption lines of other gases which follow the same mapping rules in the same interval still fluctuate. For each gas, there are two sorted pieces of CPS, and values of the sorted absorption coefficients are considerably larger than those of the unsorted pieces (note the logarithmic scale in absorption), except for the last portion of $H_2O$. Therefore, the major characteristics of gaseous absorption have been caught. For the unsorted portions, the fluctuations are limited to within a certain range; using the average values does not much affect the accuracy.

The absorption coefficient of gases is used in the infrared radiative transfer equation (IRTE), which is used to describe the atmospheric radiative transfer process. The plane-parallel IRTE describing the monochromatic radiation at frequency $v$ through a medium is given by (the subscript $v$ is omitted for the convenience of writing):

$$\mu \frac{dI(\tau, \mu)}{d\tau} = I(\tau, \mu) - J(\tau, \mu), \tag{13}$$

where $I$, $J$, $\tau$ and $\mu$ are the radiance, the source function, optical depth, the cosine of the zenith angle, respectively. For a non-scattering atmosphere, the source function of thermal radiation is

$$J(\tau, \mu) = B_T(\tau), \tag{14}$$

where $B_T(\tau)$ is the Planck function at the temperature. The solution of the radiative transfer equation (Equation (13)) is analytically obtained. Then, the solution is incorporated into this gaseous absorption scheme by AMCKD to construct the forward radiative transfer. In this model, the brightness temperatures at TOA under clear sky atmosphere can be simulated. The thermal emissions of the ground are obtained from the MODIS data [21], and the thermal emissions of the ocean are set as 0.99.

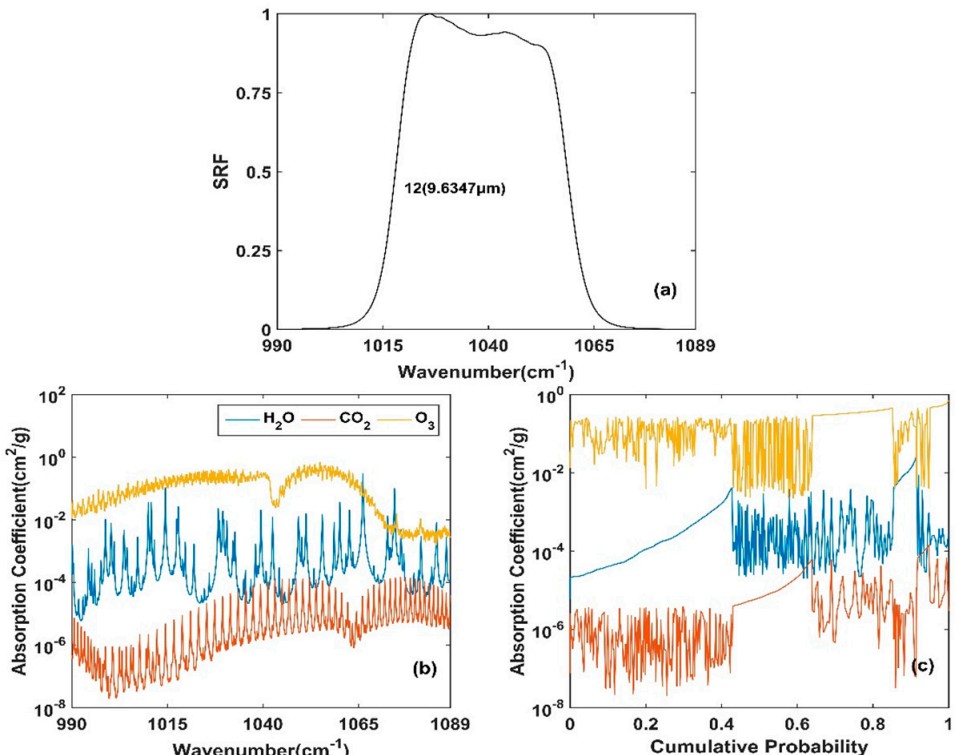

**Figure 1.** The application of AMCKD to 9.6347 μm channel of AHI. (**a**) The instrument SRF of AHI at the 9.6347 μm channel. (**b**) Gaseous absorption coefficients of $H_2O$, $CO_2$, and $O_3$, as the function of wavenumber. (**c**) Absorption coefficients in CPS with SRF after sorted by AMCKD.

## 3. Application and Results

In this section, the accuracy of AMCKD is evaluated under two types of condition. The first is under the standard atmospheric profiles. The benchmark calculations are based on LBLRTM. RTTOV is also considered in this comparison. The spectral resolution of LBLRTM is set as 0.001 cm$^{-1}$ for all the infrared channels. The LBLRTM [22,23] is used to calculate the absorption coefficient from HITRAN2012 molecular spectroscopic database [24], and the algorithm of MT_CKD_2.8 is used to deal with the water continuum absorption [25–27].

Another is under real atmospheric conditions. The simulations of brightness temperatures at TOA by AMCKD are compared to the observation results from the Advanced Himawari Imager (AHI) on board Himawari-8 and Medium Resolution Spectral Imager (MERSI) on board Fengyun-3D. The simulated area is 105° E–120° E with 25° N–40° N on December 20, 2017, at 6:00 UTC. The atmospheric input profile (atmospheric pressure, temperature, specific humidity, and ozone mass mixing ratio) are from the ERA-Interim reanalysis data [28–30]. As shown in Figure 2, there are only a few clouds in the considered area. We remove the cloudy area in the following calculation for focusing only on the gaseous transmission. The aerosol is not considered, because the aerosol infrared forcing is very weak, about one order of magnitude smaller than that in the solar spectral range.

The brightness temperature is calculated by:

$$T_B = \frac{hc}{K \ln(\frac{2hc^2}{I(0,\mu_s)\lambda^5} + 1)\lambda} \tag{15}$$

where $h$ is the Planck's constant, $K$ is the Boltzmann's constant and $\lambda$ is the wavelength of band center, $I(0,\mu_s)$ is the intensity at the top of the atmosphere observed by (AHI) and MERSI at the cosine of viewing zenith angle $\mu_s$.

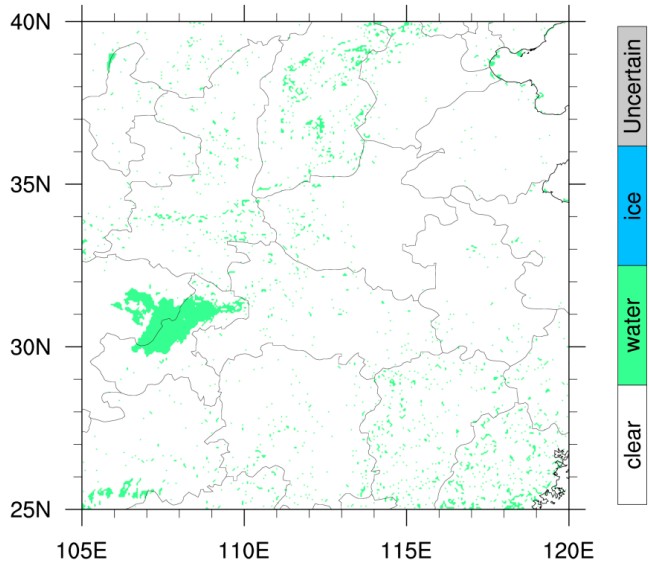

**Figure 2.** Cloud locations in the simulation area.

### 3.1. Himawari-8 AHI Channels

The bright temperatures at TOA are calculated by the forward radiative transfer algorithm with AMCKD and LBLRTM for gaseous transmission, respectively. Three standard atmospheric profiles of mid-latitude summer (MLS), sub-arctic winter (SAW) and tropical (TRO) are considered [31]. Also, the results of RTTOV are presented.

Table 1 shows the brightness temperature at TOA calculated by LBLRTM, AMCKD, and RTTOV at 45° satellite zenith angle under three atmospheric profiles. The largest absolute error of AMCKD is 0.44 K for the 7.341 μm channel under the MLS and TRO profiles. In most cases, the error of AMCKD is smaller than the NEDT at ST, while the error of RTTOV is larger than the NEDT at ST. Generally, it is necessary to use AMCKD to improve the forward radiative transfer simulation.

**Table 1.** Brightness temperatures at TOA for longwave channels calculated by LBLRTM, AMCKD, and RTTOV. The numbers shown in parentheses are the absolute errors (K) of brightness temperatures for longwave channels between each scheme and LBLRTM. The instrument noise equivalent temperature (NEDT) at scene temperature (ST) [32], indicating the tolerable noise range, is used as well in the comparisons. The satellite zenith angle is 45°. The mid-latitude summer (MLS), tropical (TRO) and sub-arctic winter (SAW) profiles are considered.

| Center Wavelength (NEDT @ ST) | | MLS | SAW | TRO |
|---|---|---|---|---|
| 7.341 μm (0.32 K @ 240 K) | LBLRTM | 258.17 | 244.44 | 259.32 |
| | AMCKD | 257.73 (−0.44) | 244.59 (0.15) | 259.76 (0.44) |
| | RTTOV | 257.57 (−0.60) | 244.29 (−0.15) | 258.81 (−0.51) |
| 8.5905 μm (0.1 K @ 300 K) | LBLRTM | 288.61 | 255.82 | 292.37 |
| | AMCKD | 288.94 (0.33) | 255.87 (0.05) | 292.58 (0.21) |
| | RTTOV | 288.30 (−0.31) | 255.42 (−0.40) | 292.16 (−0.21) |
| 9.6347 μm (0.1 K @ 300 K) | LBLRTM | 259.85 | 231.09 | 269.56 |
| | AMCKD | 259.61 (−0.24) | 230.90 (−0.19) | 269.21 (−0.25) |
| | RTTOV | 259.57 (−0.28) | 230.69 (−0.40) | 268.97 (−0.59) |
| 10.4029 μm (0.1 K @ 300 K) | LBLRTM | 291.35 | 256.95 | 295.21 |
| | AMCKD | 291.45 (0.10) | 256.96 (0.01) | 295.30 (0.09) |
| | RTTOV | 290.99 (−0.46) | 256.69 (−0.26) | 294.91 (−0.30) |
| 11.2432 μm (0.1 K @ 300 K) | LBLRTM | 290.91 | 256.79 | 294.27 |
| | AMCKD | 290.91 (−0.00) | 256.79 (−0.00) | 294.29 (0.02) |
| | RTTOV | 290.71 (−0.20) | 256.76 (−0.03) | 294.06 (−0.21) |
| 12.3828 μm (0.1 K @ 300 K) | LBLRTM | 287.85 | 256.16 | 290.45 |
| | AMCKD | 287.80 (−0.05) | 256.18 (0.02) | 290.40 (−0.05) |
| | RTTOV | 287.90 (0.05) | 256.37 (0.21) | 290.45 (0.00) |
| 13.2844 μm (0.3 K @ 300 K) | LBLRTM | 272.89 | 248.29 | 274.96 |
| | AMCKD | 273.08 (0.19) | 248.30 (0.01) | 275.03 (0.07) |
| | RTTOV | 272.65 (−0.24) | 248.29 (0.00) | 274.64 (−0.32) |

The brightness temperatures at different viewing zenith angles simulated by LBLRTM, AMCKD, and RTTOV are shown in Figure 3. We consider the 7.3471 μm and 11.2432 μm channels of AHI because the first one is an important water vapor band and the latter is located in the atmospheric window region. Both channels show relatively accurate results for RTTOV in Table 1. For the 7.3471 μm channel, the absolute errors of AMCKD against LBLRTM are bounded by 0.35 K. The error decreases with the increase of cosine of viewing zenith angles. RTTOV is obviously inferior to AMCKD, with errors larger than 0.4 K at all viewing zenith angles. For the 11.2432 μm channel, the maximum absolute error AMCKD is less than 0.03 K, while the errors of RTTOV are more than doubled. Moreover, all errors of AMCKD are located within NEDT, but the errors of RTTOV are beyond the ranges of NEDT. The results of Table 1 and Figure 3 indicate that AMCKD is superior to RTTOV under the standard atmosphere conditions.

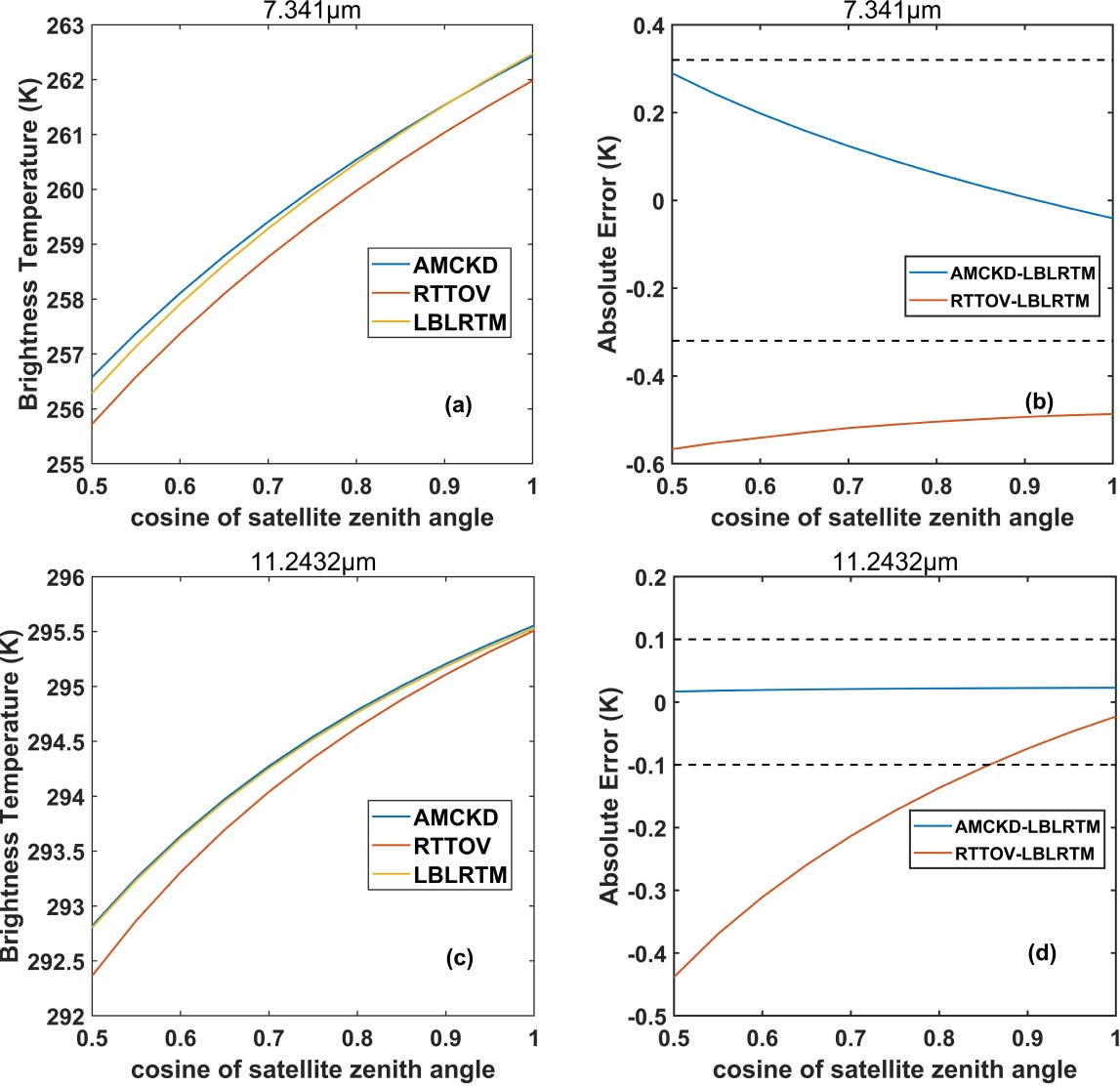

**Figure 3.** Left column (**a**,**c**): the brightness temperatures of AHI vs the cosine of viewing zenith angles for the 7.3471 μm channel (**a**) and the 11.2432 μm channel (**c**); right column (**b**,**d**): the differences of AMCKD–LBLRTM and RTTOV–LBLRTM. The dash lines are NEDTs.

Furthermore, we examine the accuracy of AMCKD in real atmospheric profiles. The simulated area is shown in Figure 2. The simulated time is on December 20, 2017, at 6:00 UTC. For AMCKD and RTTOV, the atmospheric input profiles (e.g., temperature, water vapor, and ozone) are from the

ERA-Interim reanalysis data [28–30]. In Figure 4, the comparison of simulated brightness temperature and observations of AHI in the 7.3471 μm and 11.2432 μm channels are shown, and the joint probability distribution function (PDF) (‰) is used. Most PDF results are distributed around the 1:1 standard line (simulations are equal to observations) for both AMCKD and RTTOV, which indicates that most of the simulation results are close to the observations. For the 7.3471 μm channel, both AMCKD and RTTOV are very accurate, with the absolute error mainly being less than 1 K. For the 11.2432 μm channel, in the range between 280 K and 290 K, the errors of AMCKD can reach 5 K, and the results of RTTOV are similar to those of AMCKD. The accuracies of both AMCKD and RTTOV become lower with the real atmospheric profile compared to those in the standard atmospheric profiles shown in Figure 3. In the case of standard atmospheric profiles, the calculations of the benchmark LBLRTM and simulated AMCKD and RTTOV are based on the same atmospheric profiles and the same surface thermal emissions. However, in the real atmosphere case, the atmospheric input profiles for AMCKD and RTTOV are based on reanalysis data of ERA-Interim, and the surface thermal emissions are from MODIS data, which is the grid domain averaged results. The grid domain of the reanalysis data is in coarse resolution of 0.125° × 0.125° (about 14 km × 14 km), and time resolution of the thermal emissions data is 8 days, while the sensor of an observation detects a single location of the atmosphere at a specific time. The inconsistency in input atmospheric and surface information data could lead to bias in results.

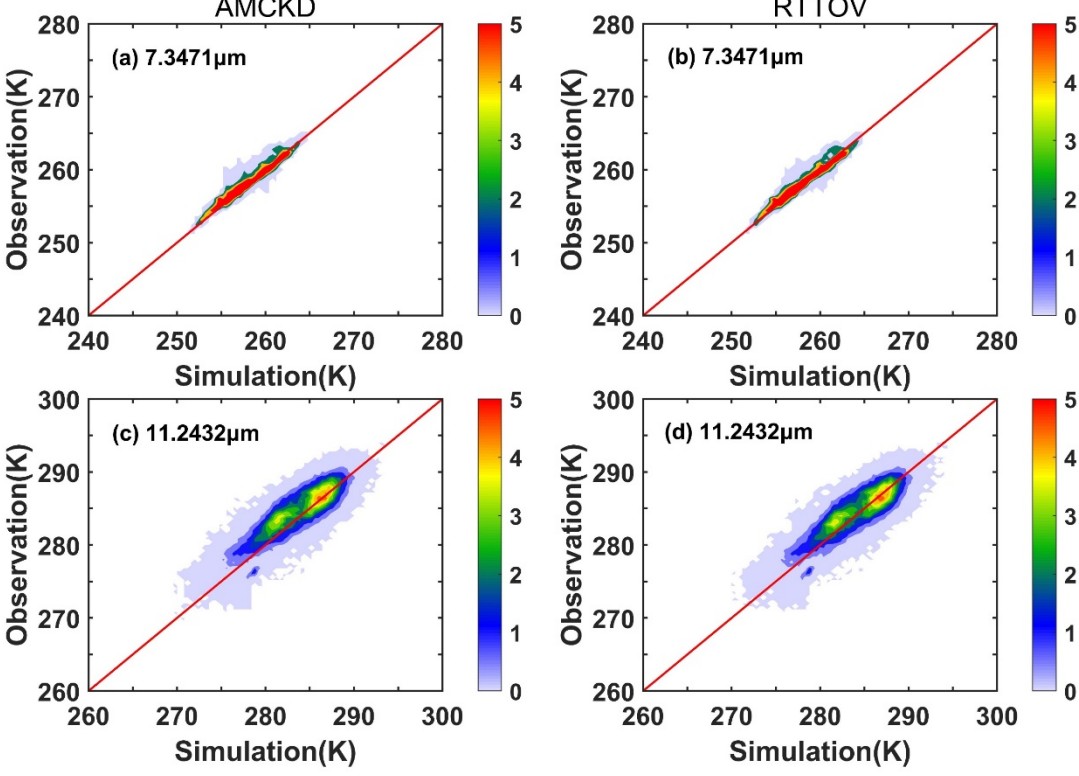

**Figure 4.** The joint probability distribution function (PDF) (‰) of simulated brightness temperature by AMCKD (**a,c**) and RTTOV (**b,d**) and observations of AHI for the 7.3471 μm (**a,b**) and 11.2432 μm (**c,d**) channels with a clear sky.

Figure 5 shows the comparisons of brightness temperature at channels of 7.3471 μm and 11.2432 μm between observed and simulated results. Both simulations by AMCKD and RTTOV are smoother than the observations, because of the lower spatial resolution of the input ERA-Interim data compared to that of satellite observations. Most of the simulated results are consistent with observations, especially for 7.3471 μm channel. For 11.2432 μm channel, apparent differences between simulations

and observations are shown in some areas. In this case, the strong absorbing gases, like $H_2O$, can efficaciously rely on the atmospheric profile data.

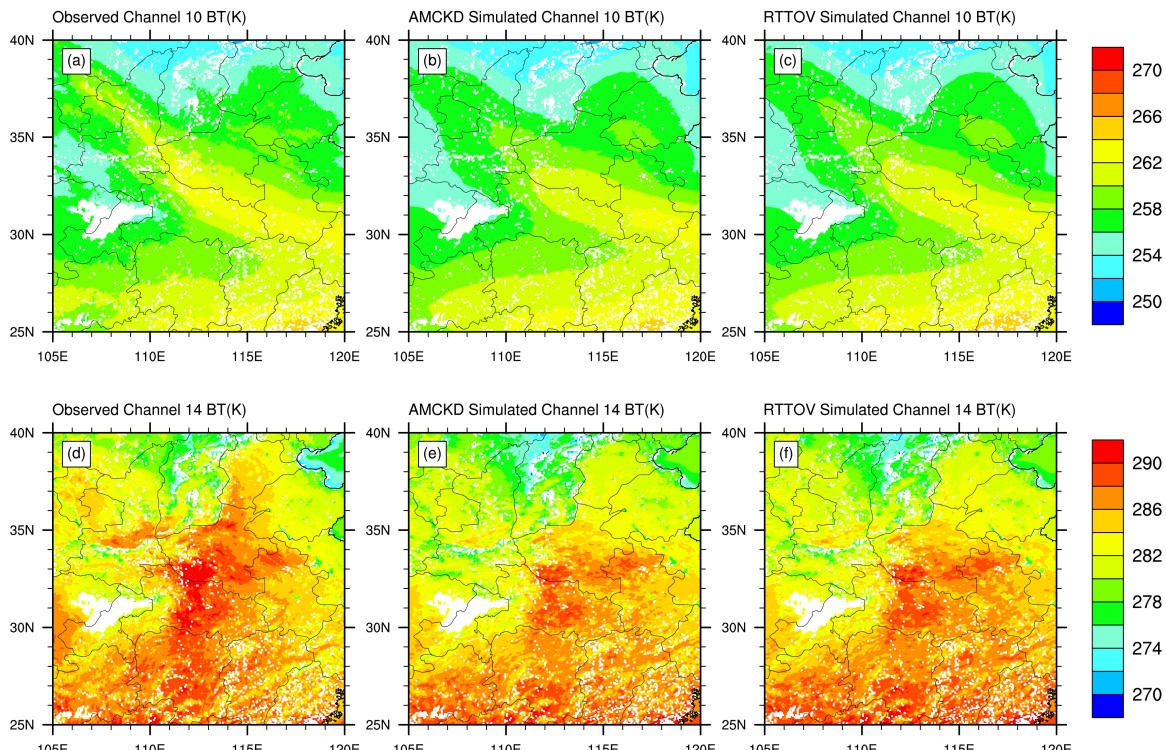

**Figure 5.** Comparison between simulated brightness temperature by AMCKD (**b**,**e**), RTTOV (**c**,**f**) and observations (**a**,**c**) of AHI 7.3471 μm (channel 10) (**a**–**c**) and 11.2432 μm (channel 14) (**d**–**f**) channels with a clear sky. The blank area is the cloudy area.

## 3.2. Fengyun-3D MERSI Channels

In the following, we examine the accuracy of AMCKD for MERSI in real atmospheric profiles, under the clear sky condition. In Figure 6, the comparison of simulated brightness temperature and observations of MERSI in 4.0459 μm and 10.7139 μm channels are considered by using the joint probability distribution function (PDF) (‰). The domain area and time are the same ast he Himawari-8 AHI case. At 4.0459 μm channel, an essential band for water vapor and $CO_2$, the PDF distributions of AMCKD are closer to the 1:1 standard line compared to RTTOV; thus, the absolute errors and the standard deviations of AMCKD are smaller than that of RTTOV. The 10.7139 μm channel mainly contains water vapor and $O_3$; the results of AMCKD and RTTOV are very similar.

The distributions of brightness temperature for the 4.0459 μm and 10.7139 μm channels are presented in Figure 7. Compared to the results of observations, the results of AMCKD are considerably more accurate than RTTOV in almost all regions. The 4.0459 μm channel contains water vapor and $CO_2$. The better results of AMCKD means that the gaseous overlap is more appropriately treated in AMCKD. For the 10.7139 μm channel, the results of AMCKD and RTTOV are similar, which is consistent with the results of Figure 6.

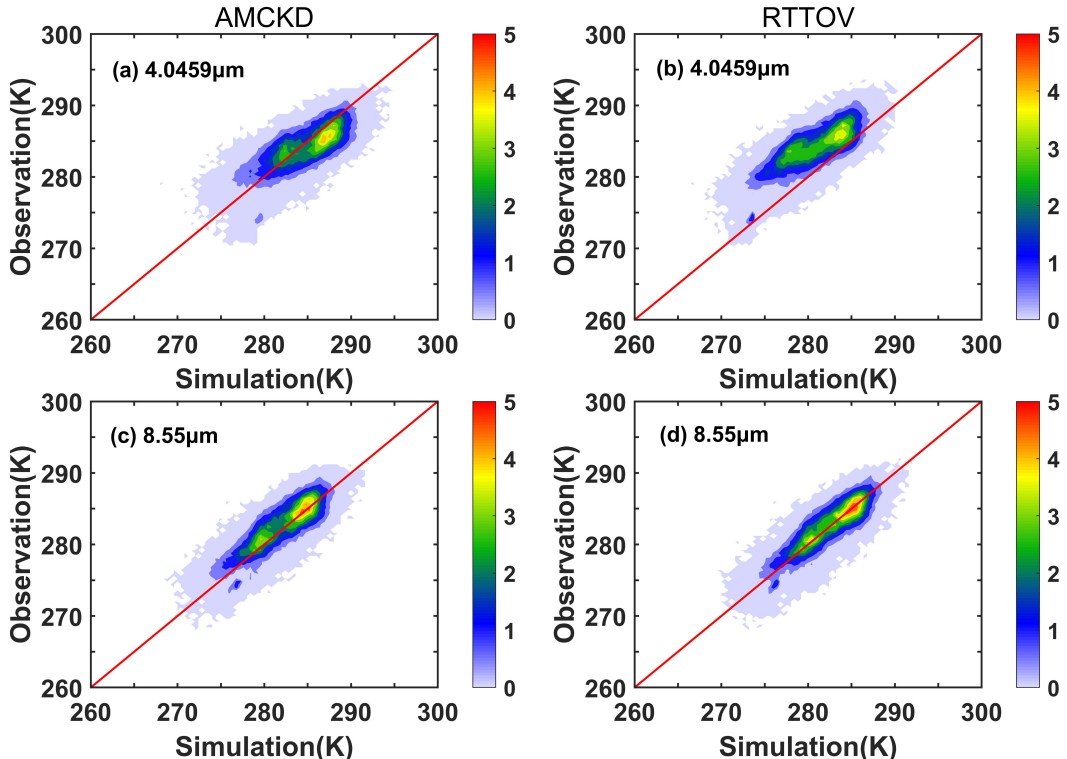

**Figure 6.** The joint probability distribution function (PDF) (‰) of simulated brightness temperature by AMCKD (**a**,**c**) and RTTOV (**b**,**d**) and observations of MERSI for the 4.0459 μm (**a**,**b**) and 10.7139 μm (**c**,**d**) channels with a clear sky.

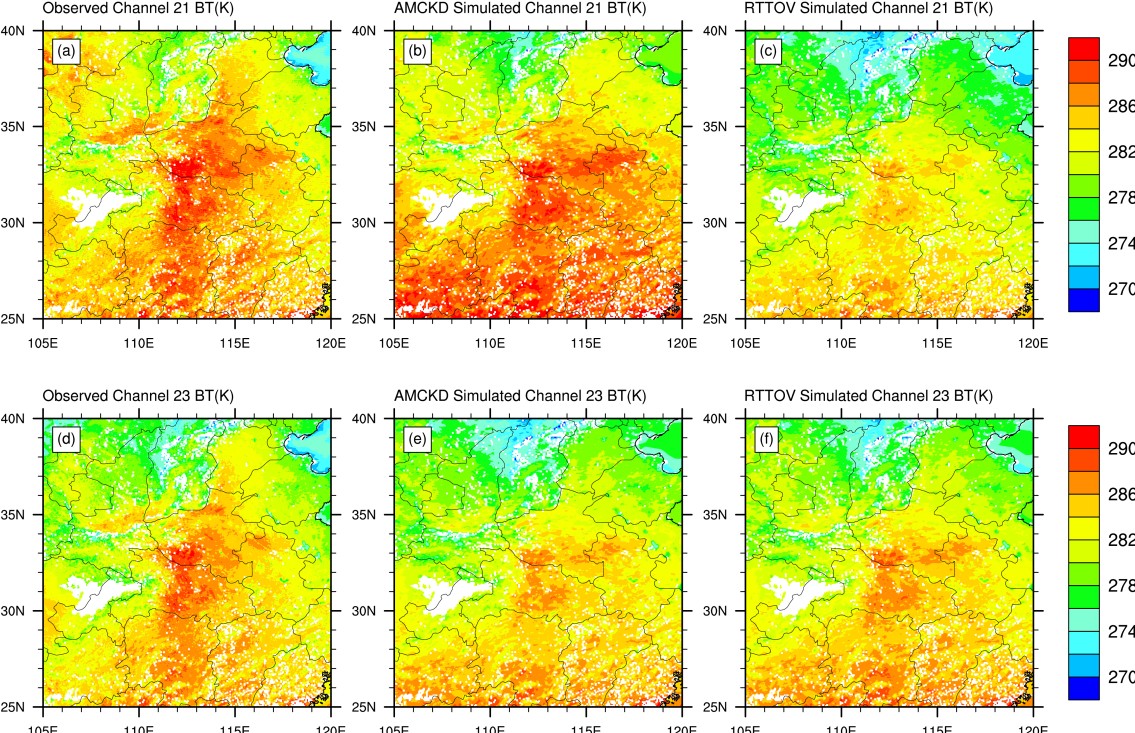

**Figure 7.** Comparison between simulated brightness temperature by AMCKD (**b**,**e**) and RTTOV (**c**,**f**) and observations (**a**,**d**) of MERSI 4.0459 μm (channel 21) (**a**–**c**) and 10.7139 μm (channel 23) (**d**–**f**) channels with a clear sky. The blank area is the cloudy area.

*3.3. Computational Time*

Table 2 compares the runtimes of LBLRTM, AMCKD, and RTTOV, where the nine-channels of AHI are simulated. The spectral resolution of LBLRTM is set as 0.001 cm$^{-1}$, and the atmosphere profile is divided into 60 layers. In all simulations, AMCKD is about 1400 times faster than LBLRTM but slightly slower than RTTOV.

**Table 2.** Comparison of the runtime of LBLRTM, AMCKD, and RTTOV.

|         | LBLRTM   | AMCKD | RTTOV |
| ------- | -------- | ----- | ----- |
| Runtime | 1723.186 | 1.182 | 1     |

## 4. Summary

AMCKD is an extended CKD method that is beneficial for handling gaseous overlap. In this study, AMCKD method has been extended to the application of the forward radiative transfer model for remote sensing, and the results are compared to LBLRTM. Under the standard atmospheric profile, the absolute errors of AMCKD in all longwave channels of AHI and MERSI are bounded by 0.44K compared to the benchmark results of LBLRTM, which is more accurate than the results of RTTOV. Under the real atmospheric profile condition, the errors of AMCKD increase, because the input data from ERA-Interim averaged by coarse grid domain can cause the bias of simulation compared with the observation of the AHI or MERSI, which focuses on a single location. However, in the most considered cases, the accuracy of AMCKD is higher than RTTOV, while the efficiency of AMCKD is slightly slower than RTTOV. For the weakness of AMCKD, it requires more effort in building the algorithm compared to the band-mean models, which need to try to choose different interval divisions and gases in each interval.

**Author Contributions:** Conceptualization, F.Z.; Methodology, J.L. and M.Z.; Software, M.Z., J.L. and W.L.; Validation, M.Z., W.L. and D.D.; Formal analysis, M.Z. and J.L.; Investigation, F.Z. and M.Z.; Resources, M.Z. and W.L.; Data curation, F.Z.; Writing—original draft preparation, F.Z., M.W. and W.L.; Writing—review and editing, F.Z., M.W., J.L., Y.-N.S. and K.W.; Visualization, M.Z. and W.L.; Supervision, F.Z.; Project administration, F.Z.; Funding acquisition, F.Z.

**Funding:** This research is supported by the National Key R&D Program of China (2018YFC1507002) and National Natural Science Foundation of China (41675003 and 41675056).

**Acknowledgments:** The authors appreciate two anonymous reviewers for their constructive comments. AMCKD code is available online at https://github.com/zmw941015/ckd.git.

**Conflicts of Interest:** The authors declare no conflict of interest.

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
