# Peer review of "Alternate Mapping Correlated k-Distribution Method for Infrared Radiative Transfer Forward Simulation"

_remotesensing, doi:10.3390/rs11090994_

Round 1

Reviewer 1 Report

The Authors addressed all my comments appropriately.

I believe, the paper now can be accepted for publication.

Author Response

We would like to thank Reviewer 1 for considering the article “the paper now can be accepted for publication”, and  we would like to thank Reviewer 1 for your valuable time, and for providing constructive remarks.

Reviewer 2 Report

The paper has greatly improved and I think now is ready for publication. I have one more final request concerning Tab. 1. In this table NEDT has to be shown together with the scene temperature at which it has been calculated, As an example authors could adopt the usage  e.g., 0.21 K @ 280 K, putting the correct scene temperature. Please don't forget that NEDT is depending on scene temperature!

Author Response

This manuscript is a resubmission of an earlier submission. The following is a list of the peer review reports and author responses from that submission.

Round 1

Reviewer 1 Report

The paper suffers from several severe pitfalls and must undergo major revision before acceptance.

First and foremost: independent reproduction of the reported results (isn't it one of criteria of scientific research ? ) is not possible. There is absolutely no instructions on how these results can be reproduced, there is no link to the Author’s software, results in Tables 1 and 3 cannot be reproduced due to insufficient input data: e.g. any aerosol model was used? The Authors refer to standard atmospheric profiles, but standards tend to change over time: there is no reference in line 156, before Table 1 to what exactly profile was used. I agree, the modeled results can be computed without aerosols and clouds, but I can hardly believe no aerosols and clouds got into the Himawari field of view. What version of HITRAN (or another database) was used for spectral simulations?

Second, throughout the paper, the Authors refer to a small error they get with their method. However, what are the instrument requirements? I see no reference and no discussion of the accuracy requirements. It makes no sense to “over converge” and spend time for that. The runtime, by the way, is not discussed at all.

Third: nothing comes for free – what are the negative features of the Authors approach?

Fourth: I left confused with the authors numerical simulation of thermal radiation as a function of view direction. As I know, clime models deal with energy balance – so they deal with the solution of RT equation integrated over view zenith and azimuth (and sometimes over solar angle – for atmospheric spherical albedo). It is the aerosol and cloud science that bother about angular distribution of reflected and transmitted light – but no aerosol model is discussed in the paper. I would replace the results in Tables with fluxes integrated over spectral range, or at least add a few new results of the kind.

Fifth and final of the major comments: Table 2 is discussed in the text, but where is it?

*** 

Other my comments along the paper (not in order of importance):

  Line 27: “radiation transmission” – to me sounds like radiation transmitted through atmosphere towards the ground.

  Section 1 “Introduction” discusses only correlated-k distribution (CKD) and, obviously, line-by-line (LBL). The authors only say in line 32 that CKD is “the most popular scheme to replace LBL”. Are there any methods beyond CKD, LBL, and the presented AMCKD? Google Scholar quickly leads me to A.Lyapustin’s 2003: “Interpolation and Profile Correction (IPC) method for shortwave radiative transfer in spectral intervals of gaseous absorption” J. Atm.Sci. - not sure if it is good or bad, but that method is not referenced. I also see no reference or discussion of the Continuum problem – if it is ignored, please say so, if it is included that what version? It is of course insufficient to just give a reference – analysis of pros and contras of different methods is essential.

  Line 48: I believe, “In climate simulation models …” should be a new paragraph.

  Line 66: numerical integration with equal intervals is selected here, while line 83 uses Gauss-Legendre quadrature. Which method is better and why? Why different methods are used?

  Eq.(5): I see a square-symbol before k(g)

  Lines 128-140: Why the surface reflection and thermal emission of the ground are ignored?

  Line 138: What version of DISORT? A weblink to disort, in addition to literature reference, is necessary – or please explain how the code was obtained. Also, DISORT is a monochromatic code – did the Authors create an interface for DISORT? What language was used, what is the input, etc?

  Same line 138: delta-256 – does 256 refer to the number of streams? I can not check it because no reference is given. If yes, for what reason the authors used 256 streams (usually, per hemisphere – so it is 512 total)? This high order is sometimes used for clouds and coarse aerosol fraction to get accurate angular distribution. I doubt the authors need that for clean atmosphere (Rayleigh – 6-8 streams are sufficient). Otherwise, please clarify.

  Line 143: What is “ideal” atmospheric profile?

  Line 145: LBLRMT – please check MT or TM?

  Line 152: No reference to "ERA-Interim" – what is it?

  Line 156: No reference to “standard” profiles.

  Tables 1 and 3: consider adding a blank line between results with units.

  Tables 1 and 3: How “reflectance” is defined?

  Figure 5: X-axis = cosine of **azimuth**, however cosine of **zenith** is mentioned in the text, lines 225 and 234. Why the azimuth dependence is preferable to the zenith one (or vice versa)? If it is indeed necessary to plot, the Authors should consider polar plots, where both zenith and azimuth dependence are shown.

Reviewer 2 Report

The paper is dealing with the development of a forward model for the HIMAWARI-8 and Fengyun-3D MERSI imagers. The forward model is developed within the well-known framework of correlated-k distribution method.  Although, the paper is methodologically correct, it is lacking important references, the quality of presentation is not so good, and the algorithm is lacking a serious validation.

The development of fast forward models is an active area of research. There exist both monochromatic and polychromatic models, which can greatly reduce the computational burden of Line-by-Line models. Among many others, we quote kCARTA (e.g.,    doi: 10.1109/TGRS.2002.808244), sigma-IASI (e.g. doi:10.1016/S1364-8152(02)00027-0 ) and RTTOV ( doi: 10.1256/qj.02.181.). These forward models should be quoted in the manuscript and compared to  that developed by authors.

Concerning presentation authors should clarify

1.       which LBL they use and who developed it.

2.       The accuracy in terms of error in BT makes no sense, since the error can depend on the scene temperature.

3.       They need to assess the performance of their forward model in the radiance space and compare with the radiometric noise expected for each channel. To say that an error of 0.45 K is good makes no sense, good with respect to what. The figure of 0.45 has to be properly compared with the instrument noise for that channel.

4.       Fig. 2 and Fig. 5:  what is azmith? That’s not English. Also, the dependence should be on the satellite zenith angle or Field of View. And more, why there is a dependence on the field of view angle.

5.       Fig. 4 and Fig. 7: What do you mean with observations? First, on which date? which day?  In case of real observations, I would expect to see clouds in both Fig. 4 and Fig. 7.  The authors say, that those are comparisons in clear sky. But one cannot expect clear sky for a target area extending from 25 to 40 degrees in latitude.

6.       The authors say they use DISORT, however there is no description of any aerosol model they use in the simulations. Why they need multiple scattering, if they use clear sky: no clouds, no aerosol.
